# Epithelial-to-Mesenchymal Transition Mediates Resistance to Maintenance Therapy with Vinflunine in Advanced Urothelial Cell Carcinoma

**DOI:** 10.3390/cancers13246235

**Published:** 2021-12-12

**Authors:** Albert Font, Vicenç Ruiz de Porras, Begoña P. Valderrama, Jose Luis Ramirez, Lara Nonell, José Antonio Virizuela, Urbano Anido, Aránzazu González-del-Alba, Nuria Lainez, Maria del Mar Llorente, Natalia Jiménez, Begoña Mellado, Jesus García-Donas, Joaquim Bellmunt

**Affiliations:** 1Department of Medical Oncology, Catalan Institute of Oncology, University Hospital Germans Trias i Pujol, Ctra. Can Ruti-Camí de les Escoles s/n, 08916 Badalona, Spain; afont@iconcologia.net; 2Catalan Institute of Oncology, Badalona Applied Research Group in Oncology (B·ARGO), Ctra. Can Ruti-Camí de les Escoles s/n, 08916 Badalona, Spain; vruiz@igtp.cat; 3Germans Trias i Pujol Research Institute (IGTP), Ctra. Can Ruti-Camí de les Escoles s/n, 08916 Badalona, Spain; 4Department of Medical Oncology, Hospital Universitario Virgen del Rocío, 41013 Seville, Spain; mbegona.perez.sspa@juntadeandalucia.es; 5Department of Haematology, Catalan Institute of Oncology, University Hospital Germans Trias i Pujol, Ctra. Can Ruti-Camí de les Escoles s/n, 08916 Badalona, Spain; jramirez@iconcologia.net; 6MARGenomics, Hospital del Mar Medical Research Institute (IMIM), 08003 Barcelona, Spain; laranonell@vhio.net; 7Department of Medical Oncology, Hospital Universitario Virgen de Macarena, 41009 Seville, Spain; javirizuelae@seom.org; 8Department of Medical Oncology, Complejo Hospitalario Universitario de Santiago, 15706 Santiago de Compostela, Spain; urbano.anido.herranz@sergas.es; 9Department of Medical Oncology, Hospital Universitario Puerta de Hierro-Majadahonda, 28222 Madrid, Spain; aranzazu.gonzalezalba@salud.madrid.org; 10Department of Medical Oncology, Complejo Hospitalario de Navarra, 31008 Pamplona, Spain; nuria.lainez.milagro@cfnavarra.es; 11Department of Medical Oncology, Hospital General Universitario de Elda, 03600 Alicante, Spain; llorente_marost@gva.es; 12Translational Genomics and Targeted Therapeutics in Solid Tumors Laboratory, Institut d’Investigacions Biomèdiques August Pi i Sunyer (IDIBAPS), 08036 Barcelona, Spain; najimenez@clinic.cat; 13Department of Medical Oncology, Hospital Clinic de Barcelona, IDIBAPS, University of Barcelona, 08036 Barcelona, Spain; bmellado@clinic.cat; 14Division of Medical Oncology, HM Hospitales-Centro Integral Oncológico Hospital de Madrid Clara Campal, 28050 Madrid, Spain; 15Division of Hematology and Oncology, Beth Israel Deaconess Medical Center, Boston, MA 02215, USA

**Keywords:** advanced urothelial cell carcinoma, vinflunine, chemotherapy resistance, epithelial-to-mesenchymal transition, maintenance therapy

## Abstract

**Simple Summary:**

Platinum-based chemotherapy is the first-line treatment for advanced urothelial cell carcinoma (aUCC). After first-line treatment, we previously showed that maintenance therapy with vinflunine improves progression-free survival. However, some patients are resistant to vinflunine and the specific mechanisms of resistance in aUCC are unclear. We analyzed the genomic landscape and the biological processes potentially related to vinflunine activity and found that epithelial-to-mesenchymal transition (EMT) plays a pivotal role as a resistance mechanism. In experiments with cell lines, curcumin reversed EMT and sensitized cells to vinflunine. We suggest that EMT mediates resistance to vinflunine and that the reversion of this process could enhance the effect of vinflunine in aUCC patients.

**Abstract:**

In the phase II MAJA trial, maintenance therapy with vinflunine resulted in longer progression-free survival compared to best supportive care in advanced urothelial cell carcinoma (aUCC) patients who did not progress after first-line platinum-based chemotherapy. However, despite an initial benefit observed in some patients, unequivocal resistance appears which underlying mechanisms are presently unknown. We have performed gene expression and functional enrichment analyses to shed light on the discovery of these underlying resistance mechanisms. Differential gene expression profile of eight patients with poor outcome and nine with good outcome to vinflunine administered in the MAJA trial were analyzed. RNA was isolated from tumor tissue and gene expression was assessed by microarray. Differential expression was determined with linear models for microarray data. Gene Set Enrichment Analysis (GSEA) was used for the functional classification of the genes. In vitro functional studies were performed using UCC cell lines. Hierarchical clustering showed a differential gene expression pattern between patients with good and poor outcome to vinflunine treatment. GSEA identified epithelial-to-mesenchymal transition (EMT) as the top negatively enriched hallmark in patients with good outcome. In vitro analyses showed that the polyphenol curcumin downregulated EMT markers and sensitized UCC cells to vinflunine. We conclude that EMT mediates resistance to vinflunine and suggest that the reversion of this process could enhance the effect of vinflunine in aUCC patients.

## 1. Introduction

Urothelial cell carcinoma (UCC) is the fifth most common type of cancer in developed countries [1]. Platinum-based chemotherapy is standard-of-care first-line treatment for advanced UCC (aUCC). These tumors are generally chemosensitive, and objective responses are achieved with first-line therapy in 40–70% of patients [2,3]. Nevertheless, the duration of response is limited, and after progression prognosis is generally poor, with a median overall survival of 14–16 months and <20% of long-term survivors [4].

Based on the positive results of several phase II/III clinical trials [5,6,7], vinflunine, a third-generation semi-synthetic vinca alkaloid, was approved in 2009 by the European Medicines Agency (EMA) for the treatment of patients with aUCC who progressed after initial treatment with platinum-based chemotherapy. However, since the introduction of anti-PD-1/L1 immune checkpoint inhibitors as a second-line therapeutic option [8,9,10,11,12,13], vinflunine has been considered a third-line treatment for aUCC patients who have progressed to immunotherapy or combination chemotherapy [14].

Based on the benefits observed in second line, vinflunine was tested in the maintenance space. Maintenance therapy can be a useful treatment strategy to delay disease progression, improve quality of life, and increase overall survival both in patients with metastatic disease who are not eligible for second-line treatment but who might still expect some clinical benefit and in those who are eligible for second-line treatment but at risk of rapid disease progression [15,16]. In a previous randomized, controlled, open-label, phase II trial of aUCC patients with controlled disease after first-line cisplatin and gemcitabine, we demonstrated that maintenance therapy with vinflunine plus best supportive care resulted in significantly longer progression-free survival than best supportive care alone, with no unexpected long-term adverse effects [12,17].

Like other microtubule-targeting drugs, vinflunine inhibits microtubule dynamics by binding to tubulin dimers and destabilizing microtubules, thereby blocking cellular mitosis in the metaphase and leading to apoptosis [18,19]. Vinflunine has shown anti-proliferative, anti-angiogenic and anti-metastatic properties in vitro and in vivo [20], with a low toxicity profile compared to other anti-microtubule agents [21,22]. Although several common resistance mechanisms to anti-microtubule agents, such as P-glycoprotein (P-gp) drug efflux pump overexpression and β-tubulin alterations, have been related to vinflunine resistance [22], the specific mechanisms underlying this resistance in aUCC patients remain unclear.

In order to identify potential mechanisms of resistance to vinflunine, we have conducted a differential gene expression analysis and a pathway analysis using GSEA including epithelial-to-mesenchymal transition (EMT) and the IL-6/JAK/STAT3 pathway, between patients with good and poor outcome included in our vinflunine maintenance trial [12,17]. The overarching aim was to develop new therapeutic approaches to overcome vinflunine resistance through an adequate understanding of the mechanisms underlying this resistance.

## 2. Materials and Methods

### 2.1. Patients’ Clinicopathological Baseline Characteristics

From April 2012 to January 2015, 88 patients with aUCC received first-line chemotherapy with four to six cycles of cisplatin and gemcitabine (carboplatin permitted after cycle four), with response or stable disease, according to the Response Evaluation Criteria in Solid Tumors (RECIST), version 1.1 at 21 hospitals and member institutions of the Spanish Oncology Genitourinary Group (SOGUG). Patients had an Eastern Cooperative Oncology Group (ECOG) performance status of 1, age ≥ 75 yrs, previous pelvic radiotherapy, or creatinine clearance < 60 mL/min. Patients were randomized to receive second-line vinflunine (every 21 days as a 20-min intravenous infusion at 320 mg/m^2^ or at 280 mg/m^2^) plus best supportive care or best supportive care alone. One patient was lost immediately to follow-up and nine discontinued treatment due to toxicity or patient decision [12,17].

We classified patients with good or poor outcome to vinflunine treatment based on the number of vinflunine cycles received before progression. Patients who progressed after more than 12 cycles were considered patients with good outcome (15 patients), while those patients who progressed before receiving four cycles (19 patients) were considered patients with poor outcome. Sufficient formalin-fixed, paraffin-embedded (FFPE) tumor tissue for the current study was available from nine patients with good outcome and eight with poor outcome to vinflunine treatment. Clinicopathological baseline characteristics were comparable between the two groups (Table 1).

In order to identify potential genetic determinants of vinflunine resistance, we conducted a four-part investigation. First, gene expression levels were determined by microarray analysis, differential expression was determined with linear models for microarray data, and Gene Set Enrichment Analysis (GSEA) was used for the functional classification of the genes. Next, 10 genes found to be highly differential expressed between patients with good and poor outcome to vinflunine treatment were analyzed by qRT-PCR analysis. We then performed a functional enrichment analysis to examine the biological processes potentially involved in vinflunine resistance. Finally, we conducted in vitro functional studies in UCC cell lines to determine cell viability and the cytotoxicity of vinflunine with and without curcumin (a well-known inhibitor of EMT) (see Graphical Abstract).

### 2.2. FFPE-Tissue RNA Extraction

For the gene expression analysis, total RNA was isolated from 10-μm sections of FFPE tumors fixed on positively charged slides. After selection of tumor areas by macrodissection, cells were lysed and sheared by sonication with an S2 (Covaris Inc., Wobrun, MA, USA) and RNA was purified using the truXTRAC FFPE microTube RNA Kit (Covaris Inc., Wobrun, MA, USA) following the manufacturer’s instructions, eluted in a final volume of 30 µL, and immediately quantified using the Qubit 3 Fluorometer (Thermo Fisher Scientific, Waltham, MA, USA).

### 2.3. Microarray Gene Expression Profiling

Gene expression microarrays were performed with Clariom S Arrays (Affymetrix/ThermoFisher) at Josep Carreras Leukemia Research Institute (Badalona, Spain). Fifty ng of total RNA from each sample were processed according to the GeneChip™ 3′ IVT Pico Kit user guide. The Affymetrix^®^ 450 fluidics station and GeneChip^®^ Scanner 3000 7G were used to wash, stain and scan the arrays.

Statistical packages included (R v3.3.2), Bioconductor [23], and the Comprehensive R Archive Network (CRAN) (http://cran.r-project.org/ accessed on 1 November 2021) were used for statistical analyses. For normalization, we used the Robust Multi-array Average algorithm (RMA) [24] in the Affymetrix package [25]. Differentially expressed genes were identified with the limma package [26]. The sva package (R v3.40.0) was used to estimate batch effects and other artifacts and adjust them into the limma model. Genes were considered differentially expressed when *p* ≤ 0.05 and the fold change |FC| was >1.5.

For gene annotation, we used the annotation file of Affymetrix with the Clariom_S_Human array (NetAffx na36) and the UCSC database (Nov. 2016 hg38, GRCh38). The Affymetrix annotation file obtained the location of each probeset (start, stop, strand and chromosome), which was then used to map genes with the same coordinates in the UCSC database.

To study whether there were differences between cell subtypes in both patient cohorts, we applied the EPIC deconvolution method to the linear RMA normalized data. This method is available through the R immunedeconv package [27]. Statistical differences between patients with good and poor outcomes were assessed with the Student’s *t*-test (Wilcoxon rank sum test).

### 2.4. Functional Enrichment Analysis

GSEA was used to retrieve functional annotation on the genes identified as differentially expressed [28]. This method links the microarray expression profile with gene sets available in the Molecular Signatures Database (MSigDB, v 6.1) (http://www.broadinstitute.org/gsea/msigdb/index.jsp, accessed on 1 November 2021). Gene sets in MSigDB are grouped in eight collections, three of which were used in the present study: hallmark gene sets, C2 curated gene sets, and C5 GO biological process. GSEA determines whether the members of a gene set tend to aggregate toward the top or the bottom of a ranked list of genes. GSEA software with the option Pre-Ranked Analysis was used (http://www.broadinstitute.org/gsea/index.jsp, accessed on 1 November 2021) to rank genes using the p-values obtained in the limma analysis. The ranked list of genes was generated using the −log(*p.val*) × signFC for each gene. Gene sets were considered significant when *p* ≤ 0.05 and FDR *q* value < 0.05.

### 2.5. Quantitative RT-PCR (qRT-PCR)

Genes selected for qRT-PCR analysis were *IGFBP3*, *IGF2*, *CXCL8*, *CCDC80*, *S100A9*, *TM4SF1, SCIN, CXorf57, EMX2, KMKN, CDH2, FN1* and *ZEB1*. qRT-PCR was performed as previously described [29]. Briefly, retrotranscription was performed with moloney murine leukemia virus (MMLV) reverse transcriptase (Thermo Fisher Scientific, Waltham, MA, USA). Template cDNA was amplified using commercial TaqMan gene expression assays and TaqMan Universal Master Mix (Applied Biosystems, Foster City, USA).

Relative gene expression quantification was calculated according to the comparative ΔCt method (ΔCt = Ct [gene] − Ctul [endogenous]), as previously described [30], with β-actin (ACTB) (Thermo Fisher Scientific) as the endogenous control. Student’s *t*-test was used to assess statistical differences between patients with good and poor outcome. *p*-values were corrected using the FDR method [31].

### 2.6. Cell Culture

The HT1376 and T24 UCC cell lines [32], kindly provided by Dr. Francesc Xavier Real (Spanish National Cancer Research Centre, Madrid, Spain), were used for the in vitro functional studies to determine cell viability and the cytotoxicity of vinflunine and curcumin alone and in combination. The in vitro studies were performed as previously described [33,34]. Briefly, the HT1376 cells were cultured in MEM (Thermo Fisher Scientific, Waltham, MA, USA) and the T24 cells were cultured in McCoy’s 5A (Thermo Fisher Scientific, Waltham, MA, USA), supplemented with 10% of heat-inactivated FCS (Reactiva, 08004 Barcelona, Spain), 400 units/mL penicillin, and 40 μg/mL gentamicin (Thermo Fisher Scientific, Waltham, MA, USA). Cell lines were cultured at 37 °C in an atmosphere of 5% CO2, periodically tested for mycoplasma contamination, and authenticated by short tandem repeat profiling.

Vinflunine (MedChemExpress, Monmouth Junction, NJ, USA) and curcumin (Sigma Aldrich, St. Louis, MO, USA) were prepared in dimethylsulfoxide and ethanol absolute, respectively

### 2.7. Cell Viability Assay

Drug cytotoxicity was assessed by the 3-(4,5-dimethylthiazol-2-yl) 2,5-diphenyltetrazolium bromide (MTT) assay (Thermo Fisher Scientific, Waltham, MA, USA). T24 urothelial cancer cells were seeded in a 96-well microtiter plates (Thermo Fisher Scientific, Waltham, MA, USA) at 5000 cells per well and allowed to attach. Medium containing different drug concentrations of vinflunine, curcumin and their combination was added after 24h. After 72h of treatment, MTT was then added and doses for 10–90% of cell viability were determined by the median-effect line method.

### 2.8. Analysis of Combined Drug Effects

The cytotoxicity of the combined drugs was assessed with serial dilution of both drugs at 1/8, 1/4, 1/2, 1, 2 and 4 of the individual IC50 values by MTT test. Fractional survival was then calculated by dividing the number of cells in drug-treated plates by the number of cells in control plates. The synergistic effect of the combined treatments was analyzed by calculating the Combination Index (CI), using Compusyn Software (Combosyn Inc) based on Chou and Talalay method, as previously described [34]. According to this method, synergism is indicated by CI < 1, antagonism by CI > 1, and additivity by CI = 1.

### 2.9. Colony-Formation Assay

To evaluate the cytotoxicity of vinflunine, curcumin and their combination, colony-formation assays were also performed as previously described [33,34]. Briefly, a serial dilution of T24 cells was made in order to seed 500 cells/well in a six-well plate (Thermo Fisher Scientific, Waltham, MA, USA), and cells were left 24 h to adhere. The following day, different dilutions of the drugs were added for 72 h. Cells were left a total of 10 days in culture for colonies to form performing regular medium changes. Cells were subsequently washed with PBS, fixed with a methanol/acetic acid (3:1) solution for 10 min and stained with a solution of crystal violet (0.5%) for 10 min. After staining, cells were washed with PBS and colonies were counted with ImageJ software.

### 2.10. Western Blotting

Western blotting was performed as previously described [33,34]. Briefly, cells were washed with PBS and homogenized in a radio immunoprecipitation assay (RIPA) plus buffer. Then, we determined the protein concentration with the DC Protein Assay (Bio-Rad Laboratories, Inc., Richmond, CA, USA). For the western blot, 50 μg of protein were loaded in a 10% sodium dodecyl sulfate-polyacrylamide gel electrophoresis (SDS-PAGE) gels (Thermo Fisher Scientific, Waltham, MA, USA) and subjected to electrophoresis. Afterwards, proteins were transferred onto a polyvinylidene difluoride (PVDF) membrane (Bio-Rad Laboratories, Inc., Richmond, CA, USA) by wet transfer. The membranes were blocked for 2 h with Odyssey blocking buffer (LICOR Biosciences, Lincoln, NE, USA) and then incubated overnight at 4 °C with specific primary antibodies against E-cadherin (Cell Signaling, Danvers, MA, USA, Ref #3195; 1:1000), N-cadherin (Cell Signaling, Ref #13116; 1:1000), Fibronectin (Cell Signaling, Ref # #26836; 1:1000), Vimentin (Abcam, Cambridge, MA, USA, ab92547; 1:1000), ZEB1 (Cell Signaling, Ref #70512; 1:1000) and β-actin (Sigma-Aldrich, #T6074, 1:2000). Membranes were incubated with IRDye rabbit and mouse secondary antibodies (1:10,000) (LICOR Biosciences, Lincoln, NE, USA) and scanned and analyzed on the Odyssey imaging system (LICOR Biosciences, Lincoln, NE, USA). Band signal was quantified with the build-in software. Each band was referenced to either β-actin band from the same sample.

### 2.11. Apoptosis Assay

Apoptosis was determined by using FITC Annexin V Apoptosis Detection Kit I (BD Pharmingen) following the manufacturer’s instructions in a FACS Canto II flow cytometer (Becton Dickinson Immunocytometry System), as previously described [34].

### 2.12. Statistical Analysis

Patients’ baseline characteristics were compared using a chi-square and Mann–Whitney U test with SPSS v.19 software.

In vitro data are reported as mean ± SEM of at least three independent experiments. The statistical analysis was performed with Graphpad Prism V.4 software. Statistical differences between IC50s were assessed with dose-response curves, non-linear regression analysis, and F-test. Different experimental conditions were compared with the Student’s *t*-test.

## 3. Results

### 3.1. Differential Gene Expression Patterns between Patients with Good and Poor Outcome to Vinflunine Treatment

Transcriptional profiling by microarray revealed 31 genes differentially expressed between patients with good and poor outcome to vinflunine treatment. Eighteen were downregulated and 13 upregulated in patients with good outcome (Figure 1).

We then selected the ten downregulated and the ten upregulated genes with the greatest FC between patients with good and poor outcome (Table 2).

We then selected 10 of these genes for qRT-PCR analysis, based on their previously described role in UCC and/or response to anti-microtubule agents. Six of these genes were downregulated in patients with good outcome (*IGFBP3*, *IGF2*, *CCDC80*, *CXCL8*, *S100A9* and *TM4SF1*) and four were upregulated (*SCIN*, *CXorf57*, *EMX2* and *DMKN*). However, the qRT-PCR analysis detected no significant differences in the expression of any of the genes between patients with good and poor outcome to vinflunine treatment, although there was a trend towards downregulation of *TM4SF1* (*p* = 0.057) and *IGFBP3* (*p* = 0.065) in patients with good outcome (Figure 2).

### 3.2. Epithelial-to-Mesenchymal Transition (EMT) Pathway Mediate Vinflunine Resistance in aUCC Patients

The functional enrichment analysis of the biological processes potentially involved in vinflunine resistance in aUCC patients identified EMT (*p* < 0.001; FDR *q* < 0.001) and the IL-6/JAK/STAT3 pathway (*p* < 0.001; FDR *q* = 0.0016) as the most negatively enriched hallmarks in patients with good outcome (Table 3; Figure 3a, b), while the G2M checkpoint (*p* = 0.017; FDR *q* = 0.019) was positively enriched in these patients (Table 3; Figure 3c).

In order to demonstrate that the identified hallmarks are contributed by the tumor and not by cancer-associated fibroblasts (CAFs) or other immune cell populations, such as tumor-associated macrophages (TAMs), we applied the EPIC deconvolution method to the linear RMA normalized data and found no statistically significant differences in the levels of TAMs, CAFs or other lymphocyte populations between patients with good and poor outcome to vinflunine treatment, suggesting that the identified hallmarks are indeed contributed by the tumor (Appendix A).

Since these results suggested a role for EMT in resistance to vinflunine in aUCC patients, we hypothesized that the combination of vinflunine with an EMT inhibitor could be an appropriate strategy to enhance vinflunine sensitivity. We therefore proceeded to test this hypothesis in vitro.

### 3.3. Downregulation of EMT Markers Enhances Vinflunine Sensitivity in UCC Cell Lines

Cells undergoing EMT display decreased expression of epithelial genes, such as E-cadherin, and increased expression of mesenchymal genes, such as N-cadherin, Vimentin and Fibronectin [35]. In fact, a crucial change that occurs during EMT is the “cadherin switch”, in which the normal expression of E-cadherin is replaced by the abnormal expression of N-cadherin [36]. In two UCC cell lines, T24 and HT1376, we analyzed the basal protein expression levels of the EMT markers E-cadherin, N-cadherin, Fibronectin and Vimentin, as well as the transcription factor Zinc-finger E-box-binding homeobox 1 (ZEB1), a key inducer of EMT, at 48, 72 and 96 h post-cell seeding. T24 cells expressed the mesenchymal proteins N-cadherin and Fibronectin, but not the epithelial marker E-cadherin. In contrast, in HT1376 cells, only E-cadherin expression was detected by Western blot. The transcriptional repressor of E-cadherin expression, ZEB1, was only expressed in the T24 cells (Figure 4a and Appendix A). In line with previous reports [37], vimentin expression was not detected in either cell line.

Morphological changes associated with EMT include loss of cell–cell contacts, the appearance of elongated mesenchymal features, and growth as single cells [38]. We observed important morphological differences between the two cell lines. In contrast to HT1376 cells, the T24 cells contained a large number of elongated, spindle-shaped fibroblast-like cells with well-developed microvilli (Figure 4b).

Since these results suggested that T24 cells present a mesenchymal-like phenotype whereas HT1376 cells have an epithelial phenotype, we selected the T24 cell line for further analysis. First, we assessed the effect of vinflunine treatment on the viability of T24 cells by treating them with increasing doses of vinflunine for 72 h. As expected, vinflunine exposure decreased cell proliferation (Figure 4c) and colony formation (Figure 4d) of T24 cells in a dose-dependent manner.

Several EMT inhibitors are known to enhance chemosensitivity [39], including curcumin (diferuloylmethane), a polyphenol that has been shown to inhibit EMT in several human tumors [40], including bladder cancer [41], thereby overcoming chemoresistance and enhancing the antiproliferative effects of conventional chemotherapy [42,43]. Based on this evidence, as well as on our previous experience with curcumin as a chemosensitizer in colorectal cancer [33,44], we investigated the effect of increasing concentrations of curcumin on T24 cell viability over 72 h. Curcumin decreased cell proliferation (Figure 5a) and colony formation (Figure 5b) of T24 cells in a dose-dependent manner. We then assessed the expression of N-cadherin, Fibronectin and ZEB1 in T24 cells after 72 h of curcumin treatment at 10 and 15 μM. As expected, curcumin decreased both protein (Figure 5c) and gene (Figure 5d) expression of these three proteinsin a dose-dependent manner, especially at 15 μM. Based on these results, we reasoned that adding curcumin to vinflunine treatment could enhance vinflunine sensitivity. Therefore, we treated T24 cells with different concentrations of concomitant vinflunine plus curcumin for 72 h. Cell viability (Figure 5e,f) and clonogenic assays (Figure 5g) showed that the addition of curcumin acted synergistically to sensitize cells to vinflunine, especially at high doses of both drugs. Furthermore, the combination treatment was associated with an increase in apoptosis compared to each of the drugs individually (Figure 5h).

## 4. Discussion

Vinflunine, a third-generation semi-synthetic vinca alkaloid, is currently a third-line treatment option for aUCC patients who have progressed to immunotherapy and/or combination chemotherapy [45]. The mechanisms mediating its cytotoxic effect are well known, but limited knowledge is available on the potential mechanisms underlying resistance. In the present study, we have performed an exploratory functional enrichment analysis through GSEA and found that EMT and the IL6/JAK/STAT3 pathway were downregulated in patients with good outcome compared to those with poor outcome to vinflunine treatment, suggesting a role for these factors on the primary resistance to vinflunine. Furthermore, the results of our in vitro analysis suggest reversing the EMT phenotype with the combination of vinflunine and curcumin (an EMT inhibitor) could overcome vinflunine resistance.

Despite differences between patients with good and poor outcome in gene expression in our microarray analysis, we were unable to detect significant differences by qRT-PCR, possibly due to the low number of samples analyzed. However, there was a clear trend towards overexpression of *TM4SF1* and *IGFBP3* in patients with poor outcome.

*TM4SF1* is upregulated in several epithelial cancers [46], including bladder cancer [47], and promotes the proliferation, migration and invasion of cancer cells by inducing EMT and cancer stemness [46,48,49]. Importantly, several studies have shown that *TM4SF1* expression is positively correlated with chemotherapy resistance in multiple cancers [46,50,51]. *IGFBP3* plays a key role in esophageal tumor progression and metastasis by facilitating EMT [52]. In addition, in metastatic prostate cancer cells, the pharmacological inhibition of *IGFBP3* enhanced response to enzalutamide, an antiandrogen therapy, through EMT reversion [53]. These promising findings in other cancers, together with our findings in the present study, lead us to suggest that the potential role of *TM4SF1* and *IGFBP3* as predictive biomarkers of response to vinflunine treatment in aUCC merits further study in a larger cohort of patients.

In line with our results, several studies have demonstrated that EMT is a key event in the development of chemoresistance in several tumors [39,54,55], including prostate [56] and bladder cancer [57]. In a previous study by our group, EMT was positively enriched in colorectal cancer cells with acquired resistance to oxaliplatin compared to their parental cell lines [33]. Interestingly, as occurs in cancer stem cells, cells undergoing EMT show an increase in drug efflux pumps and anti-apoptotic effects [39]. The important role of the overexpression of cell membrane transporter proteins, such as P-gp, in resistance to therapeutic agents [58], including vinflunine [22], is widely known.

Several signaling pathways that promote EMT are known to contribute to drug resistance [59]. For example, hyperactivation of the IL6/JAK/STAT3 pathway promotes tumor metastasis and chemoresistance via induction of EMT through the upregulation of EMT-inducing transcription factors, such as Snail, Twist-1 and ZEB1 [60]. Interestingly, both curcumin [61] and metformin, an anti-diabetic drug [62], were found to inhibit EMT by blocking the IL-6/STAT3 axis–curcumin in hepatocellular carcinoma [61] and metformin in lung cancer [62]. Curcumin inhibits EMT through the modulation of several signaling pathways [40,41], thereby overcoming chemoresistance and enhancing the antiproliferative effects of chemotherapeutic agents in preclinical models.

Loss of E-cadherin expression is a hallmark of EMT [35]. A study analyzing E-cadherin promoter by chromatin immunoprecipitation found that a repressive histone methylation mark is present at higher levels in T24 cells than in other epithelial bladder cancer cells, leading to repression of E-cadherin [57]. In the present study, we used the T24 cell line for our in vitro analyses based on its lack of E-cadherin expression as well as on the described overexpression of several EMT markers, as reported by Aparicio and colleagues [37]. In line with other studies [41], we found that levels of N-cadherin, Fibronectin and ZEB1 decreased when T24 cells were treated with curcumin. Additionally, curcumin acted synergistically to sensitize cells to vinflunine and the combination of curcumin and vinflunine led to an increase in apoptosis.

To the best of our knowledge, this is the first study to demonstrate that the combination of vinflunine with an EMT inhibitor could be a promising strategy in aUCC. However, the treatment of aUCC has improved significantly in recent years with the incorporation of immune-checkpoint inhibitors [63] as well as with targeted therapies like erdafitinib, the first anti-FGFR treatment targeting mutations/fusions in FGFR2/3 [64], and enfortumab vedotin, a fully humanized monoclonal antibody against Nectin-4 [65]. Furthermore, avelumab, an anti-PD-L1 drug, has recently demonstrated a significant benefit as maintenance therapy in aUCC [66], thereby limiting the role of vinflunine in this setting.

Nevertheless, less than 25% of patients respond to immunotherapy [63], and the benefit of targeted therapies has limited duration in a small percentage of patients. Therefore, agents like vinflunine continue to be a valuable therapeutic option in aUCC, highlighting the need to identify mechanisms of resistance, which will allow us to determine new synergistic vinflunine-based combinations. Targeting EMT is a known strategy to overcome chemotherapy resistance [39] and our in vitro findings confirm that curcumin acts through EMT reversion. These results could form the basis to translate this hypothesis to the clinic with the goal of enhancing vinflunine activity and delaying or overcoming its resistance mechanisms.

## 5. Conclusions

Our results provide confirmation that EMT mediates resistance to maintenance treatment with vinflunine in aUCC. Moreover, the findings of our in vitro analysis suggest that the combination of vinflunine with an EMT inhibitor, such as curcumin, shows potential promise as a synergistic therapy to be explored for aUCC patients.

## Figures and Tables

**Figure 1 cancers-13-06235-f001:**
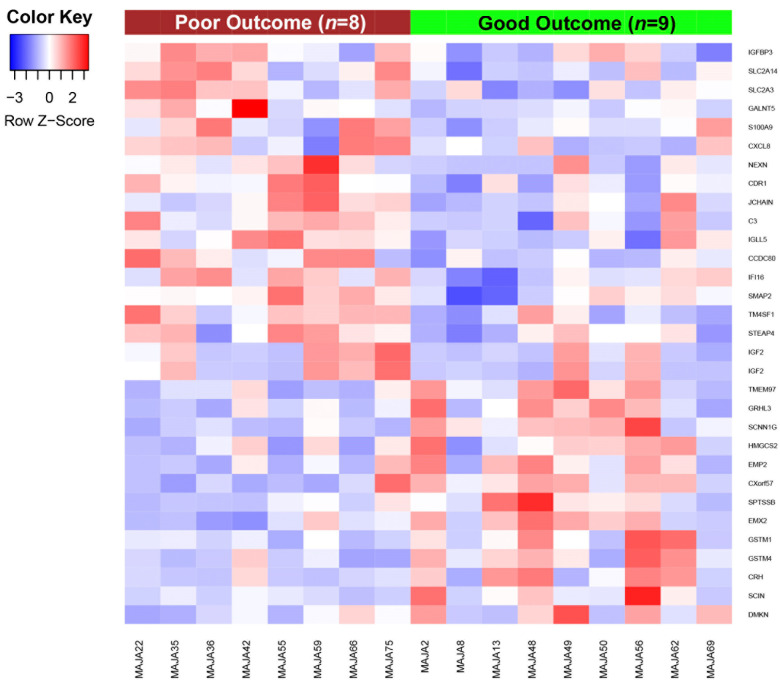
Gene expression patterns associated with vinflunine resistance in aUCC patients. Heat map depicting the normalized expression of the 31 genes differentially expressed between patients with good and poor outcome to vinflunine treatment, obtained from an adjusted linear model (|FC| > 1.5 and *p* ≤ 0.05). *IGF2* appears twice in the heatmap as two different transcripts of this gene were identified in the microarray analysis.

**Figure 2 cancers-13-06235-f002:**
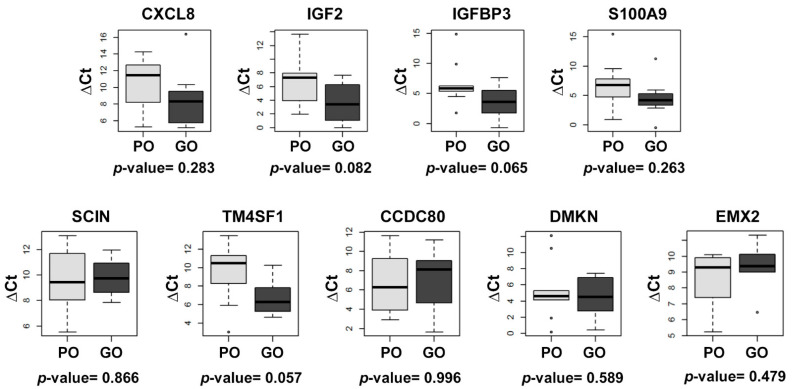
Box plots depicting the qRT-PCR relative expression of the genes differentially expressed in patients with good versus poor outcome to vinflunine treatment. β-actin (ACTB) was used as the endogenous gene. *CXorf57* expression was not detected by qRT-PCR. ΔCt: (ΔCt = Ct [gene] − Ct [endogenous]); PO: poor outcome; GO: good outcome.

**Figure 3 cancers-13-06235-f003:**
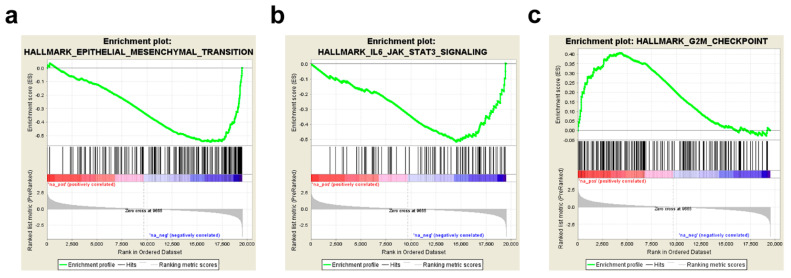
GSEA of the biological processes involved in vinflunine resistance in aUCC. GSEA plot for (**a**) EMT, (**b**) IL6/JAK/STAT3 and (**c**) G2M checkpoint.

**Figure 4 cancers-13-06235-f004:**
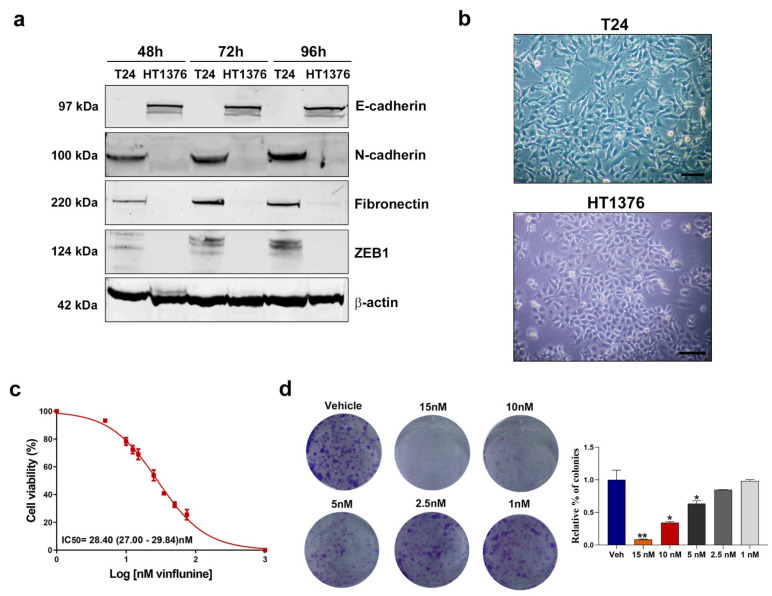
Basal protein expression levels of EMT markers in UCC cells and the effect of vinflunine on T24 cell proliferation. (**a**) Western blot analysis of E-cadherin, N-cadherin, Fibronectin and ZEB1 in T24 and HT1376 cells at 48, 72 and 96 h post-cell seeding. Beta-actin was used as endogenous control. (**b**) Phase-contrast microscopy images of the T24 and HT1376 cell lines. Scale bar: 50 µm.(**c**) Dose–response curve for T24 cells after vinflunine treatment at 0–100 nM for 72 h (mean ± SEM). IC50 value is shown as mean (95% CI). (**d**) Representative colony assay images (left) and bar graph (right) representing the percentage (mean ± SEM) of colonies in T24 cells after 72 h of vinflunine treatment at the indicated doses. * *p* < 0.05 and ** *p* < 0.01 relative to vehicle (veh) condition. All results were obtained from at least three independent experiments.

**Figure 5 cancers-13-06235-f005:**
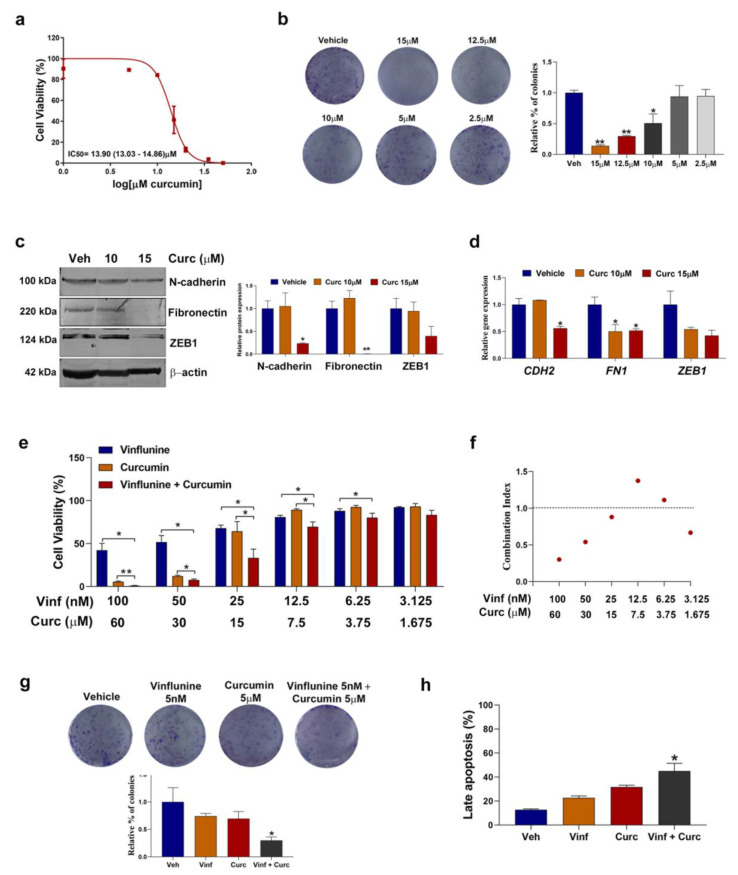
Curcumin treatment downregulated EMT markers and enhanced vinflunine sensitivity in T24 cells. (**a**) Dose–response curve for T24 cells after curcumin treatment at 0–50 µM for 72 h (mean ± SEM). IC50 value is shown as mean (95% CI). (**b**) Representative colony assay images (left) and bar graph (right) representing the percentage (mean ± SEM) of colonies in T24 cells after curcumin treatment for 72 h at the indicated doses. * *p* < 0.05 and ** *p* < 0.01 relative to vehicle. (**c**) Western blot analysis (left) and graphic representation (right) of N-cadherin, Fibronectin and ZEB1 in T24 cells after curcumin treatment for 72 h. Beta-actin was used as endogenous control. * *p* < 0.05 and ** *p* < 0.01 relative to the vehicle. (**d**) Bar graph illustrating relative gene expression levels (mean ± SEM) of N-cadherin (*CDH2*), Fibronectin (*FN1*) and *ZEB1* after 72 h curcumin treatment at the indicated doses. Gene expression levels of β–actin (*ACTB*) were used as endogenous control. * *p*-value < 0.05 relative to vehicle condition. (**e**) Bar graphs representing mean ± SEM percentage of cell viability after treatment with vinflunine, curcumin or the concomitant combination for 72 h at the indicated doses in T24 cells. * *p* < 0.05 and ** *p* < 0.01 relative to the indicated treatment (**f**) Dot plot representing combination index values calculated for each dose of the combination treatment. (**g**) Representative colony assay images (top) and bar graph (down) representing the percentage (mean ± SEM) of colonies in T24 cells after treatment with vinflunine, curcumin or their combination for 72 h at the indicated doses. * *p* < 0.05 relative to vinflunine treatment. (**h**) Bar graph representing the percentage (mean ± SEM) of late apoptotic cells after treatment with vinflunine, curcumin or the combination for 72 h. * *p* < 0.05 relative to vinflunine treatment. All results were obtained from at least 3 independent experiments. Veh: vehicle; Vinf: vinflunine; Curc: curcumin.

**Table 1 cancers-13-06235-t001:** Baseline characteristics of the 17 patients included in the present study.

Variables	Patients with Good Outcome (*n* = 9)	Patients with Poor Outcome (*n* = 8)	*p*-Value
Median age, yrs (range)	61 (47–74)	68 (52–80)	ns
Gender Male Female	7 2	6 2	ns
ECOG PS 0 1	5 4	3 5	ns
Hemoglobin < 10 gr/dL	2	0	ns
Liver metastases	0	2	ns
Number of poor prognostic factors 0 1 2	4 4 1	3 3 2	ns
Response to cis/gem CR PR SD	4 4 1	1 6 1	ns
Status (at last visit) Alive Dead	7 2	1 7	

ECOG: Eastern Cooperative Oncology Group; PS: performance status; cis/gem: cisplatin/gemcitabine; CR: complete response; PR: partial response; SD: stable disease; ns: non-significance.

**Table 2 cancers-13-06235-t002:** Genes with the greatest differences in expression between patients with good and poor outcome to vinflunine treatment, ranked by fold change.

Gene	Gene Description	Chromosome	FC *	*p*-Value *
Genes downregulated in patients with good outcome to vinflunine treatment				
C3+C18F2B3:D19	Complement component 3	chr19	−2.0511	0.0066
CDR1	Cerebellar degeneration related protein 1	chrX	−2.0409	0.0019
IGFBP3	Insulin like growth factor binding protein 3	chr7	−1.9304	0.0109
IGF2	Insulin-like growth factor 2	chr11	−1.9108	0.0343
CCDC80	Coiled-coil domain containing 80	chr3	−1.8935	0.0043
JCHAIN	Joining chain of multimeric IgA and IgM	chr4	−1.8610	0.0230
CXCL8	Chemokine (C-X-C motif) ligand 8	chr4	−1.8393	0.0399
S100A9	S100 calcium binding protein A9	chr1	−1.7811	0.0460
TM4SF1	Transmembrane 4 L six family member 1	chr3	−1.7805	0.0070
IGLL5	Immunoglobulin lambda-like polypeptide 5	chr22	−1.6382	0.0185
Genes upregulated in patients with good outcome to vinflunine treatment				
GRHL3	Grainyhead-like transcription factor 3	chr1	1.5581	0.0011
SCIN	Scinderin	chr7	1.5654	0.0120
CXorf57	Chromosome X open reading frame 57	chrX	1.5952	0.0066
GSTM1	Glutathione S-transferase mu 1	chr1	1.5981	0.0105
SCNN1G	Sodium channel non-voltage gated 1 gamma subunit	chr16	1.6793	0.0007
EMX2	Empty spiracles homeobox 2	chr10	1.6874	0.0014
DMKN	Dermokine	chr19	1.6882	0.0261
TMEM97	Transmembrane protein 97	chr17	1.7270	0.0052
CRH	Corticotropin releasing hormone	chr8	1.8411	0.0394
SPTSSB	Serine palmitoyltransferase small subunit B	chr3	1.8950	0.0144

* FC (fold change) and *p*-value shown for the comparison between patients with good and poor outcome to vinflunine treatment.

**Table 3 cancers-13-06235-t003:** Table showing the significant Hallmark terms negatively and positively enriched in patients with good outcome compared to those with poor outcome to vinflunine treatment, obtained by GSEA overlap.

Gene Set	No. Genes in Set	Gene Overlap	*p*-Value	FDR *q*-Value
Negatively enriched gene sets in patients with good outcome to vinflunine treatment				
EMT	196	73	0.000	0.000
IL6/JAK/STAT3 signaling	87	39	0.000	0.001
Coagulation	136	46	0.000	0.003
Allograft rejection	198	68	0.002	0.008
Interferon gamma response	196	75	0.000	0.007
Inflammatory response	197	71	0.000	0.023
Hypoxia	196	67	0.000	0.031
Angiogenesis	36	11	0.040	0.035
Complement	194	71	0.000	0.035
Kras signaling up	195	65	0.004	0.036
Myogenesis	199	57	0.006	0.047
Positively enriched gene sets in patients with good outcome to vinflunine treatment				
G2M checkpoint	197	61	0.017	0.019

## Data Availability

Microarray data have been deposited in the GEO database under accession number GSE181507. Other data presented in this study are available on request from the corresponding author.

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
