# Peer review of "Epithelial-to-Mesenchymal Transition Mediates Resistance to Maintenance Therapy with Vinflunine in Advanced Urothelial Cell Carcinoma"

_cancers, 2021, doi:10.3390/cancers13246235_

Round 1
Reviewer 1 Report
In this paper, the authors reported a transcriptomic analysis on post-chemotherapy aUCC patient sample and identified EMT as the key factor in resistance, which was further investigated in vitro. This is a very interesting study with clinical implication. From the perspective of academic criticism, several technical concerns need to be addressed to further improve the quality of this manuscript, as appended below.
Based on the Methods, the RNA samples were extracted from FFPE tissues, which contains multiple cell population, were used as the input for transcriptome analysis. Therefore, the hallmarks identified by bulk tissue analysis could be largely contributed by TAMs and CAFs. One way to address this issue would be looking into the TAM- and CAF-specific markers in the dataset to see if the populations were alternated in different groups.
Figure 1 has two IGF2.
A semi-quantitative analysis for the Western-blot should be supplied in Figure 4a and 5c.
Scale bars should be added to Figure 4b.
What is the difference between Figure 4d and Figure 5b?
A survival curve plot combining results from blank control, cells treated with different doses of Vinf and Curc should be added in Figure 5 or used to replace Figure 5a.
RT-qPCR should be performed on Curc treated cells to see if the treatment alternates the key marker expression.
Reviewer 2 Report
The work described involve the use of curcumin and a sensitizer of chemotherapy, vinflunine, in a model of bladder cancer. The authors present the role of curcumin as an inhibitor of epithelial to mesenchymal transdifferentiation, the proposed mechanism of resistance to vinflunine. Unfortunately, the work has limited novelty.
Round 2
Reviewer 1 Report
The manuscript is good for publication now.
Author Response
We sincerely thank the reviewer for his/her appreciation of our work.
Reviewer 2 Report
The authors have addressed the the reviewer issues.
Author Response

(The authors gave the same response as above.)
